# Kepler Algorithm for Large-Scale Systems of Economic Dispatch with Heat Optimization

**DOI:** 10.3390/biomimetics8080608

**Published:** 2023-12-14

**Authors:** Sultan Hassan Hakmi, Abdullah M. Shaheen, Hashim Alnami, Ghareeb Moustafa, Ahmed Ginidi

**Affiliations:** 1Electrical Engineering Department, College of Engineering, Jazan University, Jazan 45142, Saudi Arabia; shhakmi@jazanu.edu.sa (S.H.H.); halnami@jazanu.edu.sa (H.A.); 2Department of Electrical Engineering, Faculty of Engineering, Suez University, Suez P.O. Box 43221, Egypt; ahmed.ginidi@eng.suezuni.edu.eg

**Keywords:** Kepler optimization algorithm, economic dispatch, valve point loading effect, large 192-unit system, CHPUED

## Abstract

Combined Heat and Power Units Economic Dispatch (CHPUED) is a challenging non-convex optimization challenge in the power system that aims at decreasing the production cost by scheduling the heat and power generation outputs to dedicated units. In this article, a Kepler optimization algorithm (KOA) is designed and employed to handle the CHPUED issue under valve points impacts in large-scale systems. The proposed KOA is used to forecast the position and motion of planets at any given time based on Kepler’s principles of planetary motion. The large 48-unit, 96-unit, and 192-unit systems are considered in this study to manifest the superiority of the developed KOA, which reduces the fuel costs to 116,650.0870 USD/h, 234,285.2584 USD/h, and 487,145.2000 USD/h, respectively. Moreover, the dwarf mongoose optimization algorithm (DMOA), the energy valley optimizer (EVO), gray wolf optimization (GWO), and particle swarm optimization (PSO) are studied in this article in a comparative manner with the KOA when considering the 192-unit test system. For this large-scale system, the presented KOA successfully achieves improvements of 19.43%, 17.49%, 39.19%, and 62.83% compared to the DMOA, the EVO, GWO, and PSO, respectively. Furthermore, a feasibility study is conducted for the 192-unit test system, which demonstrates the superiority and robustness of the proposed KOA in obtaining all operating points between the boundaries without any violations.

## 1. Introduction

### 1.1. Motivation of the Study

Power and heat systems that are combined bear the responsibility of meeting both electrical and heating demands, which contradicts the conventional methods of electricity generation. As a result, the goal of the combined heat and power economic dispatch issue is to minimize both the total cost of producing electrical power and heat under certain operational restrictions, including the production attributes of combined power and heat (CHP) units, the electrical power, the heat balance, the production capacities of heat-only and power-only units, etc. [1]. Formally, the CHPUED problem under consideration can be represented mathematically as a non-convex constrained optimization issue. By using appropriate optimization techniques with the CHPUED issue, the ideal generation schedule for power and heat can be established. One of the primary motivations for this study is the increasing demand for efficient and sustainable power generation in large-scale systems. Also, the researchers recognize the need for advanced optimization algorithms that can handle the challenges posed by large-scale systems and can incorporate heat optimization aspects. Moreover, the inclusion of heat optimization in the study’s objectives highlights another important motivation. Heat optimization involves optimizing the utilization of waste heat generated during power generation processes, thereby enhancing the overall energy efficiency.

### 1.2. Literature Review

Economic load dispatch (ELD) is a crucial optimization problem in power systems that aims to allocate the optimal power generation among multiple generating units to meet the electricity demand at minimum cost while satisfying various operational constraints [2]. The objective of ELD is to determine the optimal power output for each generator, taking into account factors such as fuel cost, transmission losses, and system constraints. The goal is to minimize the total cost of generating electricity while ensuring that the power supply remains reliable and stable [3]. ELD considers various parameters, including the cost curves of generators, generation limits, ramp rates, transmission constraints, and renewable energies [4]. In Ref. [5], the application of an enhanced particle swarm optimizer was described for a dispatching model that takes into account various factors. These include market power sales benefits, environmental benefits from grid-connected operations, and system operation and maintenance costs. The objective of the model was to optimize the benefits of a combined system consisting of wind, photovoltaic, and concentrating solar power by optimizing its grid-connected performance. In Ref. [6], the sparrow search algorithm was utilized to address the economic load dispatch (ELD) problem in renewable integrated microgrids. The goal was to determine the optimal power output of all distributed energy sources within the microgrid, considering renewable sources, to meet the load demand at the lowest possible cost.

The initial attempts at addressing the CHPUED problem in the literature involved using deterministic methods, which required investigating the solution space over a limited number of repetitions, with the aid of a set of deterministic bounding procedures. The CHPUED problem is divided into two subproblems as indicated in Ref. [1], with the power and heat dispatches being handled separately by the Lagrangian relaxation approach. In Ref. [5], a bi-level structure with lower and upper levels was used to tackle the CHPUED issue. Additionally, in this study, the global limitations were managed at the top level, and unit generation was achieved at the lower level. Another study [6] used Benders decomposition to divide the CHPUED problem into a master problem and a sub-problem. In each iteration, the master issue was first solved to determine the heat productions, and the subproblem was then solved to produce the best power generation. Power dispatch and unit commitment issues combine to form CHPUED [7]. Thus, inner and outer Bender decompositions, in the two Benders decomposition techniques, were proposed. While the outer inner problem addressed a power dispatch, the outer master problem resolved the unit commitment problem by identifying the on/off status of each unit. Moreover, the inner decomposition ensured constraint fulfillment. Other traditional methods used to address the CHPUED problem in the literature included a branch-and-bound algorithm [8] and mixed-integer nonlinear programming [9].

The CHPUED problem becomes a non-convex problem when it is defined within the constraints of the valve point effect. Although the optimal solution is guaranteed by traditional approaches, these might not be able to solve the non-convex CHPUED issue. Metaheuristics carry out certain procedures that use their stochastic nature to investigate the solution space. Although they cannot guarantee that they will be close to the global optimum, they can nevertheless handle non-differentiable and non-convex problems and are simple to employ [10,11,12]. Harmony search (HS), differential evolution (DE), particle swarm optimization (PSO), and the genetic algorithm (GA) are some well-known examples of metaheuristics algorithms. A self-adaptive real-coded genetic algorithm was used to solve the non-convex CHPUED problem in Ref. [13]. In Ref. [14], PSO was utilized to solve the CHPUED problem in order to minimize the total generation cost. However, the application of PSO in this context has been limited to a small-scale system consisting of only six units, comprising two generators, two heat-only units, and two power-only units. Furthermore, the issue of transmission losses has not been considered, leading to certain inconsistencies. Additionally, the validation of the PSO approach has not been thoroughly discussed. The research conducted by [15] presented a novel approach to address the CHPUED problem by introducing a modified version of the bat optimization algorithm. This modified algorithm was designed to handle three distinct types of units to minimize the operating costs. In [10], an elitist variant of the cuckoo search algorithm has been applied for benchmark functions. In addition to this application, the elitist cuckoo search has been dedicated to solving the CHPUED problem. On the other side, the computational burden of 1000 iterations for the five- and seven-unit systems is considered, which represents more than three times that of other methods [16] with 300 iterations. Additionally, a new approach to constraint control with penalties is shown. Considering several cases, the proposed method’s effectiveness is tested. In Ref. [17], a novel mutation technique known as the Mühlenbein mutation is presented to increase convergence and solution time. Real-coded GA is used to solve the CHPUED problem while taking into account the transmission system losses and the valve point impact caused by the thermal units. In Ref. [18], Cauchy distribution is used to increase the PSO’s efficiency in addressing the CHPUED problem. Six separate test systems are used to validate the proposed method. Another population-based metaheuristic, PSO with time-varying acceleration coefficients, is utilized in Ref. [19] to solve the CHPUED issue, which includes generating constraints. In [20], the CHPUED problem has been treated, including renewable wind energy in order to overcome the intermittent renewable and load variations. In this study, it was solved by the metaphor-less Rao-3 algorithm, which involves the reliance on metaphors to enhance transparency, simplicity, and ease of understanding.

Along with these traditional stochastic optimization techniques, new metaheuristics have also recently been used to tackle the CHPUED problem, which represents the subject of this study. In [21], a non-convex CHPUED issue has been solved using a heap technique, which is then tested on four separate case studies with 7, 24, 48, and 96 generating units. However, the large 192-unit system has not been accounted for in this study. CHPUED, as a non-convex optimization problem, is formulated in this manuscript and solved via the Kepler optimization algorithm (KOA). In 2015 [22], an attempt to simulate the KOA in a hybrid algorithm with the gravitational search algorithm (GSA) was introduced for solving numerical benchmarks. This hybrid algorithm made use of only the first Kepler’s law with high simplification. It mimicked his law by focusing the search on the sun position by either multiplying it by a uniformly distributed pseudo-random number or by adding a distance component between the sun position and each solution. In the presented study, the KOA is developed in a complete optimization framework simulating different laws and features regarding Kepler’s concept [23]. The KOA enables a more efficient exploitation and exploration of the search space due to the candidate solutions’ (planets’) varying distances from the sun at various times. Each planet in the KOA represents a candidate solution with regard to the optimal solution (the sun), which is changed at random during the optimization process. All the practical constraints of CHPUED are considered in this paper. All operating points of different units are obtained between the boundaries without any violations.

### 1.3. Paper Contributions

The main points of this manuscript are summarized as follows:A KOA is developed and applied for the non-convex CHPUED.Three large-scale systems of 48, 96, and 192 units are considered tests for evaluating the effectiveness of the KOA.To assess the efficacy of the applied KOA, recent optimization algorithms are employed.To estimate the KOA’s superiority, comparisons are illustrated with various well-known methodologies that have been presented in the scientific literature.To demonstrate the accuracy of the proposed KOA, a feasibility study is investigated.

### 1.4. Paper Organization

The content of the paper is structured into five sections. The KOA algorithm is introduced in Section 2. The CHPUED optimization model is formulated in Section 3. Section 4 contains a detailed analysis and discussion of the numerical results for the three case studies. The final conclusions are drawn in Section 5.

## 2. CHPUED Formulation

### 2.1. Objective

The major objective of the CHPUED challenge is to reduce the fuel expenses associated with producing heat and electricity. As a result, CHPUED is described as an objective function that is subjected to a number of restrictions. Figure 1 displays a graphic representation of the economic CHPUED issue including their participants.

As the major objective of CHPUED is to lessen the whole cost of heat and power production, the whole cost target function (*WCTF*) can be described as the whole cost of the power-only, power–heat amalgamation, and heat-only units [24] as depicted in the following equation:(1)$$\mathrm{Min}\;WCTF=\sum_{i=1}^{N_{pr}}\;CT1_{i}(PR_{i}^{pr})+\sum_{j=1}^{N_{ht}}CT2_{j}(HT_{j}^{ht})+\sum_{k=1}^{N_{chp}}CT3_{k}(PR_{k}^{chp},HT_{K}^{chp})\;(\$/h)$$where the terms $CT1_{i}(PR_{i}^{pr})$ and $CT2_{j}(HT_{j}^{ht})$ demonstrate the *i*^th^ power-only and the *j*^th^ heat-only units’ cost, respectively, whilst the term $CT3_{k}(PR_{k}^{chp},HT_{k}^{chp})$ determines the *k*^th^ CHP. The symbols *N_ht_* and *N_pr_* depict the heat-only and power-only units’ number, respectively, while *N_chp_* illustrates the CHP units’ number.

The mathematical representation of three cost target functions $CT1_{i}(PR_{i}^{pr})$, $CT2_{j}(HT_{j}^{ht})$, and $CT3_{k}(PR_{k}^{chp},HT_{k}^{chp})$ can be described as follows:(1)*CT*1*_i_* of *i*^th^ power units
(2)$$CT1_{i}(PR_{i}^{pr})=\delta 1_{i}{\left(PR_{i}^{pr}\right)}^{2}+\delta 2_{i}PR_{i}^{p}+\delta 3_{i}+\left|\lambda_{i}\sin(\rho_{i}(PR_{i}^{pr_{\min}}-PR_{i}^{pr}))\right|\;(\$/h)$$(2)*CT*2*_j_* of *j*^th^ heat units
(3)$$CT2_{j}(HT_{j}^{ht})=\gamma 1_{j}{\left(HT_{j}^{ht}\right)}^{2}+\gamma 2_{j}HT_{j}^{ht}+\gamma 3_{j}\;(\$/h)$$(3)*CT*3*_k_* of *k*^th^ CHP units
(4)$$CT3_{k}(PR_{k}^{chp},HT_{k}^{chp})=\alpha 1_{k}{\left(PR_{k}^{chp}\right)}^{2}+\alpha 2_{k}PR_{k}^{chp}+\alpha 3_{k}+\alpha 4_{k}{\left(HT_{k}^{chp}\right)}^{2}+\alpha 5_{k}HT_{k}^{chp}+\alpha 6_{k}HT_{k}^{chp}PR_{k}^{chp}\;(\$/h)$$
where the elements (*δ*1*_i_*, *δ*2*_i_*, and *δ*3*_i_*) and (*γ*1*_j_*, *γ*2*_j_*, and *γ*3*_j_*) describe the *i*^th^ power-only and the *j*^th^ heat-only units’ cost coefficients, while the elements ($\alpha 1_{k},\alpha 2_{k},\alpha 3_{k},\alpha 4_{k},\alpha 5_{k}$ and $\alpha 6_{k}$) represent the *k*^th^ CHP unit’s cost coefficients. The non-differentiability and non-convexity of CHPUED, which signify the valve-point impacts, are determined by sinusoidal terms, as explained in Equation (2) [25,26].

### 2.2. Constraints

The following constraints are taken into account when minimizing the specified objective function. The balance among power generation and demand can be calculated using Equation (5):(1)Power balance constraint
(5)$$\sum_{i=1}^{N_{pr}}PR_{i}^{pr}+\sum_{k=1}^{N_{chp}}PR_{k}^{chp}=PR_{demand}$$
where $PR_{demand}$ explains the power demand.(2)Limits of power units’ capacity
(6)$$PR_{i}^{pr_{\min}}\leq PR_{i}^{pr}\leq PR_{i}^{pr_{\max}}\;i=1,\ldots ,N_{pr},$$(3)Heat balance constraint
(7)$$\sum_{i=1}^{N_{chp}}HT_{i}^{chp}+\sum_{j=1}^{N_{ht}}HT_{j}^{ht}=HT_{demand},$$
where $H_{demand}$ proves thermal demand.(4)Heat units’ generation limits
(8)$$HT_{j}^{ht_{\min}}\leq HT_{j}^{ht}\leq HT_{j}^{ht_{\max}}\;j=1,\ldots ,N_{ht},$$(5)CHP capacity limits
(9)$$PR_{i}^{chp_{\min}}(HT_{i}^{chp})\leq PR_{i}^{chp}\leq PR_{i}^{chp_{\max}}(HT_{i}^{chp})\;i=1,\ldots ,N_{chp},$$
(10)$$HT_{i}^{chp_{\min}}(PR_{i}^{chp})\leq HT_{i}^{chp_{}}\leq HT_{i}^{chp_{\max}}(PR_{i}^{chp})\;i=1,\ldots ,N_{chp},$$
where the heat and power unit boundaries are demonstrated by the superscripts “min” and “max”.

## 3. Mathematical Model of KOA

The Kepler optimization algorithm (KOA), proposed in [23], is inspired from Kepler’s laws for planetary motion. Each planet in the KOA acts as a candidate solution and can be updated at random during the optimization process in relation to the best-possible solution (the sun). A set of initial objects (possible solutions) containing stochastic orbitals is used by the KOA to begin the search process. During this phase, each object is first placed in an orbit at a random location. The KOA runs in iterations after assessing the original set’s fitness until the termination condition is satisfied. Because iteration is a term that is frequently used in solar system theory and cosmic cosmology, the term “time” is used in the present study instead of the iteration. In the following section, the KOA will be presented in six steps.

### 3.1. Step 1: Initialization Process

In this procedure, the decision parameters of an optimization issue, which are represented by a number of planets *N_p_* and called the population size, will be distributed at random in *Dim* dimensions as follows:(11)$${\vec{X}}_{i,j}(0)=r_{1}\times {\vec{X}}_{j,up}+{\vec{X}}_{j,low}(1-r_{1}),i=1:N_{p};j=1:Dim$$
where *X_i,j_* signifies the *i*^th^ candidate solution (planet); *N_p_* represents the number of candidate solutions in the search space; *X*_*j*,*low*_ and *X*_*j*,*up*_ characterize the lower and upper limits of the *j*^th^ decision parameter, respectively; *Dim* denotes the issue dimension to be enhanced; and *r* stands for a number randomly generated between 0 and 1.

### 3.2. Step 2: Calculating an Object’s Velocity

An object’s velocity is influenced by $V_{{}_{i}}(t)$ where it is in relation to the sun. In other words, a planet moves faster when it is near to the sun and slower when it is farther away. The sun’s gravity is quite powerful when an object is close to it, so the planet tries to move faster to escape being drawn toward the sun. The weak gravity of the sun will force an object’s velocity to slow down if it is distant from the sun. This behavior is mathematically described in Equation (12), which uses it to calculate an object’s velocity around the sun using the vis viva equation. This equation has two parts as shown below:(12)$$V_{{}_{i}}(t)=\left\{\begin{array}{c} ({\vec{X}}_{a}-{\vec{X}}_{i})\times r_{4}\times H+F\times {\vec{U}}_{2}\times \left(1-R_{i-norm}(t)\right)\\ \times (r_{3}{\vec{X}}_{i,up}-{\vec{X}}_{i,low}){\vec{r}}_{5}\;if\;R_{i-norm}(t)>0.5\\ (2\times r_{4}\times {\vec{X}}_{i}-{\vec{X}}_{b})\rho +({\vec{X}}_{a}-{\vec{X}}_{b})\rho^{*}\\ +F\times {\vec{U}}_{1}\times \left(1-R_{i-norm}(t)\right)\times ({\vec{X}}_{i,up}-{\vec{X}}_{i,low}){\vec{r}}_{5}\;Else \end{array}\right.$$
where
(13)$$H=\sqrt{(Ms+m_{i})\times \mu (t)\left|-\frac{1}{a_{i}(t)+\varepsilon }+\frac{2}{R_{i}(t)+\varepsilon }\right|}$$
(14)$$\rho =(r_{3}\times (1-r_{4})+r_{4})\times \vec{U}\times H$$
(15)$$\rho^{*}=(r_{3}\times (1-r_{5})+r_{5})\times (1-\vec{U})\times H$$
where *V_i_*(*t*) represents the velocity of object *i* at time *t*; *X_i_* characterizes an object *i*, whilst *F* manifests an integer number chosen randomly that belongs to the set {−1, 1}; the symbol $\vec{U}$ is a vector containing integer number randomly chosen which belongs to the set {0, 1}; *r*_1_, *r*_2_, *r*_3_, *r*_4_, and *r*_5_ denote random integer numbers uniformly distributed in the range [0, 1]; *μ*(*t*) denotes the universal gravitational constant; *X_a_* and *X_b_* signify solutions that are selected from the population at random; *Ms* and *m_i_* characterize the mass of *X_s_* and *X_i_*, respectively; *Ri*(*t*) demonstrates the distance at time *t* among the best solution (*X_s_*) and the object (*X_i_*); *ε* demonstrates a minimal value for avoiding a divide-by-zero mistake; and *a_i_* is the elliptical orbit semimajor axis at time *t* of object *i*, and it is defined by Kepler’s third law as follows:(16)$$a_{i}(t)={\left[\mu (t)\times (Ms+m_{i})\times \frac{T_{i}{}^{2}}{4\pi^{2}}\right]}^{\frac{1}{3}}\times r_{3}$$
where *T_i_* is an absolute value that is produced at random using the normal distribution to represent the orbital period of object *i*. The semimajor axis of object *i*’s elliptical orbit is considered in our proposed algorithm to steadily decrease over generations as the solutions advance toward the region where the best overall solution is most likely to be discovered. *R_i−norm_*(*t*) denotes normalizing the Euclidian distance among *X_s_* and *X_i_*; its definition is as follows.
(17)$$R_{i-norm}(t)=(R_{i}(t)-R_{\min}(t))/(R_{\max}(t)-R_{\min}(t))$$

### 3.3. Step 3: Escaping from the Local Optimum

Most solar system objects rotate on their own axes and move in an anticlockwise manner around the sun; however, certain objects move in a clockwise motion. This behavior is used by the approach suggested to leave local optimal zones. The suggested KOA simulates this behavior by employing a flag *F* that modifies the search direction such that agents have a good chance of accurately scanning the search space.

### 3.4. Step 4: Updating Objects’ Positions

As previously explained, objects have their own elliptical orbit around the sun. Objects rotate near the sun, becoming closer to it for a while and then moving farther from it. The two main parts of the proposed KOA, which are the exploitation and exploration phases, can simulate this behavior. The KOA searches for new locations near the best solutions while employing solutions close to the sun more precisely and exploring things far from the sun to locate new solutions. The fact that the objects are far from the sun throughout the exploration phase shows that the suggested method efficiently explores the whole search area.

The following equation is used to update the location of each object far from the sun in line with the preceding steps:(18)$${\vec{X}}_{i}(t+1)={\vec{X}}_{i}(t)+{\vec{V}}_{i}(t)\times F+({\vec{X}}_{s}(t)-{\vec{X}}_{i}(t))\times \vec{U}\times \left(Fg_{i}(t)+\left|r\right|\right)$$
where *X_i_*(*t +* 1) stands for the new location at time *t* + 1 of an object *I*, while *X_i_*(*t*) represents the present location of the object *i* at time *t*; *V_i_*(*t*) illustrates the velocity of object *i* that is needed to transfer to the new location, *X_s_*(*t*) manifests the best sun position which is associated with the best solution that acquires the least fitness score, and *F* is demonstrated as a flag to switch the search’s direction.

The term *Fg_i_* in the context of the Kepler optimization algorithm (KOA) represents the attraction force between the sun (*X_s_*) and any planet (*X_i_*). This force is calculated based on the universal law of gravitation and can be expressed using the following equation:(19)$$Fg_{i}(t)=r_{4}+\mu (t)\times e_{i}\times ({\vec{Mn}}_{s}\times {\vec{mn}}_{i})/({\vec{Rn}}_{i}{}^{2}+\varepsilon )$$
where *e_i_* manifests the eccentricity of a planet’s orbit, which is a value between 0 and 1 that was proposed to endow a stochastic characteristic to the KOA. Additionally, the normalized values of *m_i_* and *Ms* can be defined by *mn_i_* and *Mn_s_* that demonstrate the mass of *X_i_* and *X_s_*, respectively.

This equation is utilized to model the gravitational force between celestial bodies, specifically the sun and the planets, as part of the algorithm’s calculations. The force is an essential factor in determining the planetary motion and optimizing the trajectories of the planets within the system. By incorporating the sun’s attraction force, the KOA can simulate the gravitational interactions between celestial bodies and effectively optimize the orbits and positions of the planets in the system being studied. This enables the algorithm to provide accurate and efficient solutions for various astronomical and astrophysical problems. The normalized values of *m_i_* and *Ms* can be mathematically represented by Equations (20) and (21):(20)$$Mn_{s}=r_{2}\times (fit_{s}(t)-worst(t))/\sum_{k=1}^{N_{p}}\left(fit_{k}(t)-worst(t)\right)$$
(21)$$mn_{i}=(fit_{i}(t)-worst(t))/\sum_{k=1}^{N_{p}}\left(fit_{k}(t)-worst(t)\right)$$
where *worst*(*t*) represents the solution candidate with the highest fitness score; *fit_k_*(*t*) indicates the value of the fitness function regarding each location of the object *k* at the current time *t*.

The Euclidian distance between *X_s_* and *X_i_* can be defined by the term (*Rn_i_*), as depicted in Equation (22), which represents the normalized value of (*R_i_*).
(22)$$Rn_{i}(t)={\left\|X_{s}(t)-X_{i}(t)\right\|}_{2}=\sqrt{\left(\sum_{j=1}^{Dim}{\left(X_{s}(t)-X_{i}(t)\right)}^{2}\right)}$$

To manage search accuracy, *μ*(*t*) is defined by Equation (23) as a function that exponentially declines with time (*t*).
(23)$$\mu (t)=\mu_{o}\times e^{-(t/T_{\max})\gamma }$$
where *μ_o_* is an initial value; *γ* is a constant; and *T_max_* and *t* demonstrate the maximum iterations’ number and current iteration number, respectively.

### 3.5. Step 5: Updating Distance with the Sun

The normal distance behavior, which normally fluctuates with time between the planets and the sun, is simulated to further enhance the exploitation and exploration operators of planets. The KOA will concentrate on optimizing the exploitation operator when planets are near the sun and the exploration operator when the sun is far away. This principle is represented mathematically as follows:(24)$${\vec{X}}_{i}(t+1)={\vec{U}}_{1}\times {\vec{X}}_{i}(t)+\left(1-{\vec{U}}_{1}\right)\times \left({\vec{X}}_{i}(t)+{\vec{X}}_{a}(t)+{\vec{X}}_{s})/3+\frac{1}{e^{\left(r\times \left(1+r_{4}\times (a_{2}-1)\right)\right)}}\times \left(\left({\vec{X}}_{i}(t)+{\vec{X}}_{a}(t)+{\vec{X}}_{s})/3-{\vec{X}}_{b}(t\right)\right)\right)$$
where *a*_2_ defines a cyclic controlling parameter as manifested in Equation (25):(25)$$a_{2}=-(1+t/T_{\max})$$

### 3.6. Step 6: Elitism

This stage employs an elitist method in order to guarantee the optimal placements for the sun and the planets. Equation (26) provides a summary of this process. Figure 2 describes in detail the flowchart of KOA.
(26)$${\vec{X}}_{i,new}(t+1)=\left\{\begin{array}{cc} {\vec{X}}_{i}(t+1) & if\;fit\left({\vec{X}}_{i}(t)\right)\geq fit\left({\vec{X}}_{i}(t+1)\right)\\ {\vec{X}}_{i}(t) & Else \end{array}\right.$$

## 4. Performance of KOA on the CHPUED Issue

In this section, the proposed KOA is tested on large 48-unit, 96-unit, and 192-unit test systems to demonstrate its efficacy and superiority when handing the CHPED issue.

### 4.1. The 48-Unit System

The system, in this instance, involves 48 total units, including 10 power–heat combination units, 12 power-only units, and 26 heat-only units. It is necessary, in this instance, to provide 2500 MWth of heat and 4700 MW of power. Additionally, the valve-point impact for power-only units is taken into account. The capacity limits of heat-only and power-only units, as well as the cost coefficients of associated units, are taken from Reference [19]. Table 1 presents the test outcomes of all units obtained using the KOA.

The output power of the power-only units (MW) is reflected by parameters between P1 and P104. P105 and P152 are the power outputs of CHP units in MW and H105 and H152 relate to heat outputs of CHP units in MWth. Additionally, H153 and H192 are the outputs of heat-only units in MWth. As can be seen in Table 1, the Sum (Hg) and Sum (Pg) values satisfy the heat and power demands of 2500 MWth and 4700 MW, respectively. As demonstrated from Table 1, the best cost value is identified by the KOA as 116,650.0870. Additionally, all results are in the feasible zone, and several individual findings are put exactly at the lower or higher bounds.

Figure 3 illustrates the suggested KOA’s convergence rates for the given system, where the curve of the proposed KOA converges quickly. The proposed KOA requires around 2400 iterations to obtain the best solution. The results show that the proposed KOA has outstanding convergence rates for the given large CHPUED system.

Table 2 presents a comprehensive comparison between the proposed KOA and various other techniques reported in the literature. The comparison is conducted on a 48-unit CHPUED test system. The techniques included in the comparison are supply–demand optimization (SDO) [27], the multi-verse algorithm (MVA) [28], the gravitational search algorithm (GSA) [29], civilized swarm optimization (CSO) incorporated Powell’s pattern search (PPS) [30], gray wolf optimization (GWO) [28], the salp swarm algorithm (SSA) [31], the GSO-based algorithm with ranger operators and modified scrounger (MGSO) [32], CPSO [19], differential evolution (DE) [28], the crow search algorithm (CSA) [28], the marine predator algorithm (MPA) [33], the jellyfish search optimizer (JFSO) [34], the manta ray foraging algorithm (MRFA) [28], and PSO with time-varying acceleration coefficients (PSO-TVAC) [19].

Table 2 clearly demonstrates that the proposed KOA outperforms all the other optimizers in terms of performance and cost. It attains the most favorable results among the compared techniques, making it the superior choice for CHPUED optimization. The KOA exhibits the lowest minimum, standard deviation, average, and worst values of 116,650.0870, 298.8796, 117,104.5447, and 117,915.5359, respectively, as indicated in Table 2. This table clearly shows that the proposed KOA has the best performance and the lowest cost of these optimizers. Moreover, this comparison validates the proposed KOA’s efficacy and superiority when used with CHPUED. Furthermore, the proposed KOA also receives the lowest minimum, standard deviation, average, and worst values of 116,650.0870, 298.8796, 117,104.5447, and 117,915.5359 $, as shown in Table 2. Consequently, the proposed KOA has greater robustness than the techniques that have been reported. Based on the best attained costs, the proposed KOA shows improvements of 3.660%, 1.982%, 0.589%, 0.864%, 3.021%, 0.180%, 2.680%, 5.404%, 0.614%, 3.286%, 5.086%, 0.615% and 0.613%, respectively, compared to the following algorithms and optimization methods: CPSO [19], PSO-TVAC [19], MRFA [28], MVA [28], SSA [31], MPA [33], GSA [29], CSA [28], MGSO [32] DE [28], GWO [28], CSO, PPS [30] and JFSO [34]. These statistics further reinforce the robustness of the proposed KOA compared to the reported techniques.

The comprehensive analysis presented in Table 2 serves to validate the efficacy and superiority of the proposed KOA specifically when applied to the CHPUED problem. Its exceptional performance, combined with the lowest cost values, establishes the KOA as the most reliable and effective optimization approach within the context of CHPUED.

### 4.2. The 96-Unit System

The system, in this instance, involves 96 total units, including 24 power–heat combination units, 52 power-only units, and 20 heat-only units. It is necessary, in this instance, to provide 5000 MWth of heat and 9400 MW of power. Additionally, the valve-point impact for power-only units is taken into account. The capacity limits of heat-only and power-only units, as well as the cost coefficients of associated units, are taken from Reference [24]. Table 3 presents the test outcomes of all units obtained using the KOA.

H1 and H24 relate to heat outputs of CHP units in MWth, P53 and P76 are the power outputs of CHP units in MW, and between H25 and H44 are the outputs of heat-only units in MWth. The output power of the power-only units (MW) is reflected by the parameters between P1 and P52. Additionally, Sum (Hg), Sum (Pg), and WCTF stand for the total heat production (MWth), total power generation (MW), and total generation costs ($) of thermal electrical and energy, respectively. As can be seen in Table 3, the Sum (Hg) and Sum (Pg) values satisfy the heat and power demands of 5000 MWth and 9400 MW, respectively. The best cost value is identified by KOA as 234,285.3. Additionally, all results are in the feasible zone, and several individual findings are put exactly at the lower or higher bounds. Additionally, the standard HBA, the standard JSA, and the proposed HBJSA effectively achieve all constraints with 100% accuracy, as illustrated in Table 3.

Figure 4 illustrates the suggested KOA’s convergence rates for the given system, where the curve of the proposed KOA converges quickly. The proposed KOA requires around 2700 iterations to obtain the best solution. The results show that the proposed KOA has outstanding convergence rates for the given large CHPUED system.

A comparison between the proposed KOA and other reported techniques is depicted in Table 4. The reported techniques that are employed for the 96-unit CHPUED test system are the whale optimization algorithm (WOA) [24], supply–demand optimization (SDO) [27], the marine predator algorithm (MPA) [33], the improved MPA (IMPA) [33], the heap technique (HT) [34], the jellyfish search optimizer (JFSO) [34], hybrid HTJFSO (HHTJFSO) [34], the manta ray foraging algorithm (MRFA) [28], weighted vertices optimization and PSO (WVO-PSO) [35], and PSO with time-varying acceleration coefficients (PSO-TVAC) [19]. This table clearly shows that the proposed KOA has the best performance and the lowest cost of these optimizers. Moreover, this comparison validates the proposed KOA’s efficacy and superiority when used with CHPUED. Furthermore, the proposed KOA also receives the lowest minimum, standard deviation, average, and worst values of 234,285.2584, 761.7006, 235,683.2917, and 236,929.2188. Consequently, the proposed KOA has greater robustness than the techniques that have been reported. Based on the best attained costs, the proposed KOA derives improvements of 0.423%, 0.349%, 0.235%, 1.030%, 0.416%, 0.853%, 2.072%, 1.588%, 2.807% and 0.811%, respectively, compared to the following algorithms and optimization techniques: JFSO [34], HT [34], HHTJFSO [34], WOA [24], IMPA [33], MPA [33], PSO-TVAC [19], WVO-PSO [35], WVO [35] and SDO [27]. These statistics further reinforce the robustness of the proposed KOA compared to the reported techniques.

The extensive examination depicted in Table 4 provides substantial evidence to support the effectiveness and superiority of the proposed Kepler optimization algorithm (KOA) in addressing the CHPUED problem. The remarkable performance of the KOA, coupled with its cost-effectiveness, solidifies its position as the most dependable and efficient optimization approach for CHPUED applications.

### 4.3. The 192-Unit System

The system, in this instance, involves 192 total units, including 48 power–heat combination units, 104 power-only units, and 40 heat-only units. It is necessary, in this instance, to provide 10,000 MWth of heat and 18,800 MW of power. Additionally, the valve-point impact for power-only units is taken into account. The capacity limits of heat-only and power-only units, as well as the cost coefficients of associated units, are taken from Reference [24].

Table 5 presents the obtained objective costs of the proposed KOA, the dwarf mongoose optimization algorithm (DMOA) [36], the energy valley optimizer (EVO) [37], GWO and PSO. The best cost value is identified by the KOA as 487,145.2. As demonstrated from Table 5, KOA acquires a value of 487,145.2, whereas the DMOA [36], the EVO [37], GWO, and PSO have values of 581,798, 572,324.8, 678,051.9, 793,224.8, respectively. According to the obtained costs, the presented KOA successfully achieves improvement of 19.43%, 17.49%, 39.19% and 62.83% compared to the DMOA, the EVO, GWO and PSO, respectively.

Also, the detailed test outcomes of all units obtained using the applied algorithms are tabulated in Appendix A. The output power of the power-only units (MW) is reflected by parameters between P1 and P104. P105 and P152 are the power outputs of CHP units in MW and H105 and H152 relate to heat outputs of CHP units in MWth. Additionally, H153 and H192 are the outputs of heat-only units in MWth. As can be seen in Table 5, the Sum (Hg) and Sum (Pg) values satisfy the heat and power demands of 10,000 MWth and 18,800 MW, respectively.

Figure 5 illustrates the convergence rates of the KOA, the DMOA, the EVO, GWO, and PSO for the given system, where the curve of the proposed KOA converges quickly. Regarding Figure 5, Table 6 presents the detailed setting of parameters for the applied algorithms. As shown in Figure 5, the proposed KOA requires around 1500 iterations to obtain the best solution. The results show that the proposed KOA has outstanding convergence rates over the DMOA, the EVO, GWO, and PSO for the given large CHPUED system. It can be seen from Figure 5 that the DMOA and EVO seem to have a line convergence characteristic from the first to the last iteration. To clarify this point, a close zooming vision is displayed for the convergence rates of each individual applied algorithm in Figure 6. As shown in Figure 6b, the DMOA does not show a line characteristic, but on the contrary, it shows a gradual decrease in the objective. Compared to the KOA in Figure 6a, the proposed KOA shows a smooth converging feature, while the DMOA convergence is unsmooth like stairs. On the other side, the EVO starts searching and finding solutions to minimize the objective. Unfortunately, it derives a straight line after approximately half of the iterations’ journey. This indicates that this method is stuck in a local minimum. This analysis illustrates the significant convergence characteristics of the proposed KOA against the DMOA, the EVO, GWO, and PSO.

To display the simulation time, Table 7 records the average time per each iteration of the proposed KOA for all cases. As shown in this table, the simulation time for the large-scale system of 192 units is 0.1399 s, where it records 0.0952 for the 48-unit system with time increased by more than 32%. The more the scalability of the case study under investigation increases, the more simulation time is required. By utilizing the well-known Big O notation, the computational complexity of the applied algorithm can be estimated by multiplying the number of design variables, number of solutions and maximum number of iterations. Based on that, the computational complexity for each case study is recorded in Table 8.

### 4.4. Feasibility Study for 192-Unit System

A feasibility study is conducted for the 192-unit test system when applying the KOA. Figure 7, Figure 8, Figure 9 and Figure 10 display the operating points related to power-only units, CHP units and heat-only units, respectively. As illustrated, all operating points are found between the boundaries of power-only, heat-only, and CHP units. These results demonstrate the effectiveness of the proposed KOA in obtaining practical feasible solutions without any violations. All results are in the feasible zone, and several individual findings are put exactly at the lower or higher bounds. Furthermore, as shown in Figure 7, Figure 8, Figure 9 and Figure 10, the proposed KOA completely and accurately satisfies all criteria.

### 4.5. Discussion

In the previous subsections, the proposed KOA is tested on large 48-unit, 96-unit, and 192-unit test systems. Different remarks are discussed, which are summarized in the following paragraphs.

Table 1 and Table 3 and Appendix A provide a comprehensive depiction of the operating points of the power-only, CHP, and heat-only units for all units in the three investigated systems. These tables effectively describe and demonstrate that the operating points of all units, which are categorized as power-only, CHP, and heat-only, are maintained within the specified limits. This serves as evidence that the proposed KOA successfully preserves the operating points within the defined boundaries.

The convergence rates of the suggested KOA are visualized in Figure 3, Figure 4 and Figure 6 for the three investigated systems. These figures vividly illustrate the high-quality and rapid response of the KOA’s convergence rates. The plots demonstrate the algorithm’s ability to quickly converge toward optimal solutions, indicating its efficiency and effectiveness.

Table 2, Table 4 and Table 5 present various comparisons between the proposed KOA and other reported techniques for the three investigated systems. The extensive analysis provided in these tables showcases the exceptional performance and cost-effectiveness of the proposed KOA when compared to alternative methods. The KOA not only achieves superior results in terms of CHPUED optimization, but it also exhibits greater robustness compared to its counterparts. These findings strongly validate the credibility and value of the proposed KOA as an efficient and reliable optimization solution for CHPUED applications.

For the large 192-unit test system, a feasibility study is conducted and analyzed using Figure 7, Figure 8, Figure 9 and Figure 10. These figures demonstrate that all operating points fall within the boundaries of power-only, heat-only, and CHP units. The results illustrate the effectiveness of the proposed KOA in obtaining practical and feasible solutions while avoiding any violations. All the obtained results lie within the feasible zone, and several individual findings precisely align with the lower or upper bounds. Furthermore, as depicted in Figure 7, Figure 8, Figure 9 and Figure 10, the proposed KOA satisfactorily fulfills all the defined criteria, demonstrating its accuracy and compliance with the given constraints.

## 5. Conclusions

### 5.1. Paper’s Findings

In this study, the KOA is developed for the non-convex CHPUED issue. Kepler’s laws of planetary motion serve as the main source of inspiration for the KOA. According to Kepler’s laws, four operators—position, gravitational force, mass, and orbital velocity—affect how planets move around the sun. To demonstrate the effectiveness of the proposed KOA methodology, three test CHPUED systems are chosen, which are 48, 96, and 192-unit systems. Additionally, new optimizers are introduced for the large 192-unit test system, which are the DMOA, the EVO, GWO, and PSO. A feasibility study is conducted, which demonstrates the superiority and robustness of the proposed KOA. Furthermore, the proposed KOA delivers the lowest overall cost values for the three test systems when compared with various well-known methodologies that have been presented in the scientific literature and the new approaches that are implemented for the first time in this research.

### 5.2. Future Works

The proposed Kepler optimization algorithm (KOA) demonstrates significant potential for effectively addressing the CHPUED problem in large-scale systems. There is room for further enhancement in the system’s performance.
One potential area of enhancement is to upgrade the model by incorporating external market signals.Integrating external factors and signals from the market can help determine the optimal dispatch scenario.The constraints of the transmission losses can be considered, which add more complexity to the study.The scope of the work can be expanded by incorporating the emission dispatch of thermal units.Considering the environmental impact and emissions of the thermal units can lead to more sustainable and environmentally friendly dispatch solutions.The integration of renewable energies should be considered for future extensions of the work, promoting a greener and more sustainable energy mix.

## Figures and Tables

**Figure 1 biomimetics-08-00608-f001:**
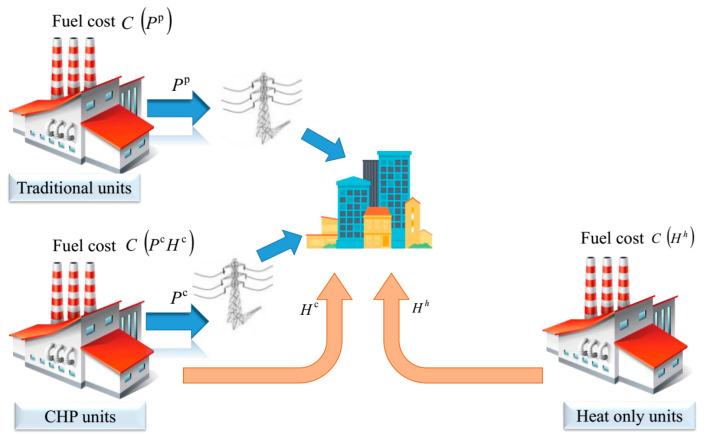
Graphic representation of economic CHPUED issue.

**Figure 2 biomimetics-08-00608-f002:**
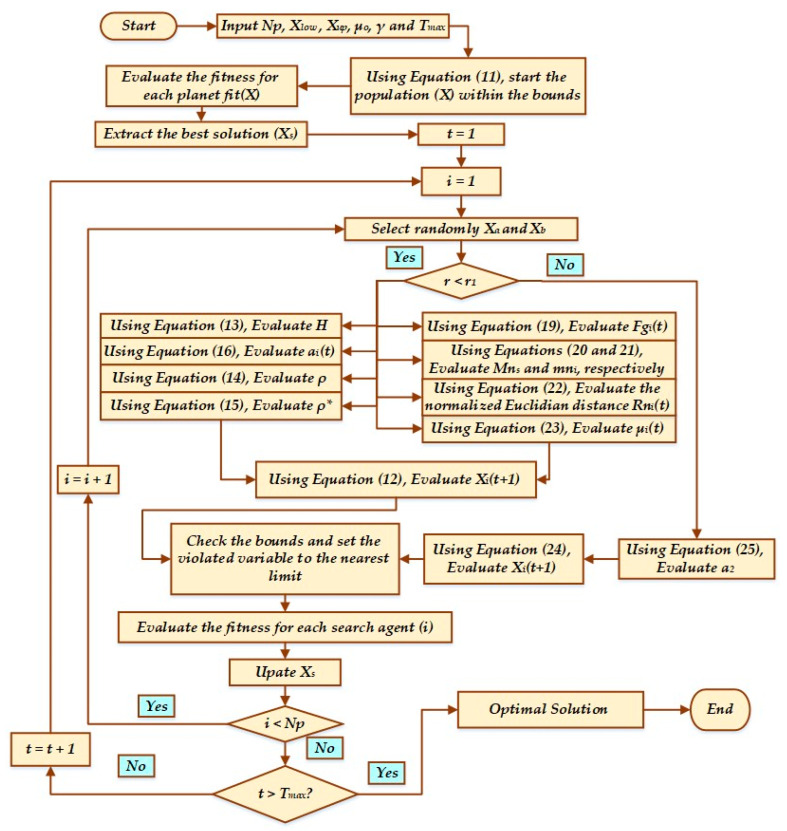
Flowchart of KOA.

**Figure 3 biomimetics-08-00608-f003:**
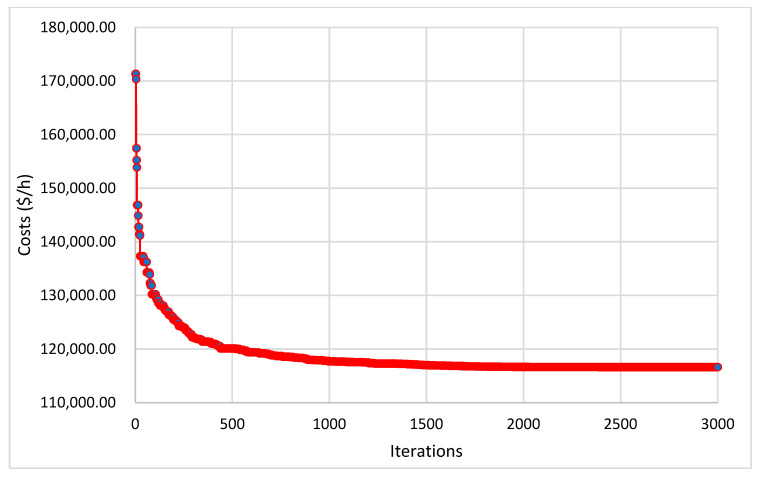
Convergence rates of the proposed KOA for 48-unit CHPUED system.

**Figure 4 biomimetics-08-00608-f004:**
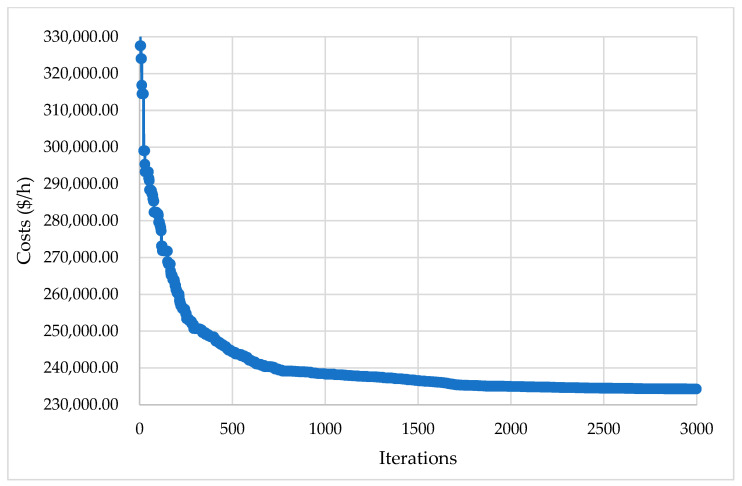
Convergence rates of the proposed KOA for 96-unit CHPUED system.

**Figure 5 biomimetics-08-00608-f005:**
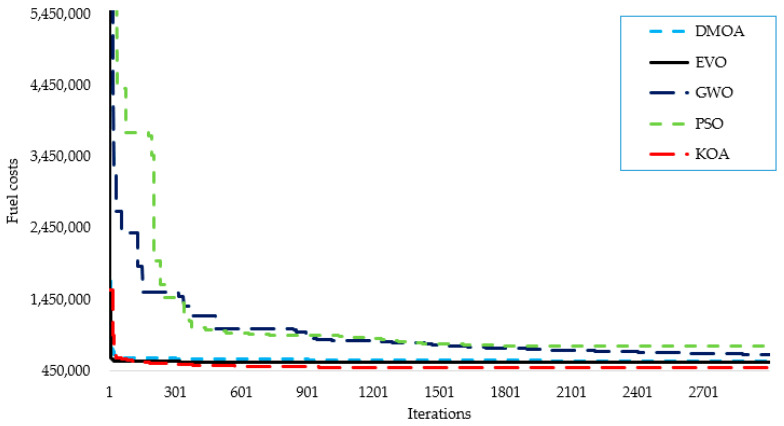
Convergence rates of the proposed KOA, DMOA, EVO, GWO, and PSO for the 192-unit CHPUED system.

**Figure 6 biomimetics-08-00608-f006:**
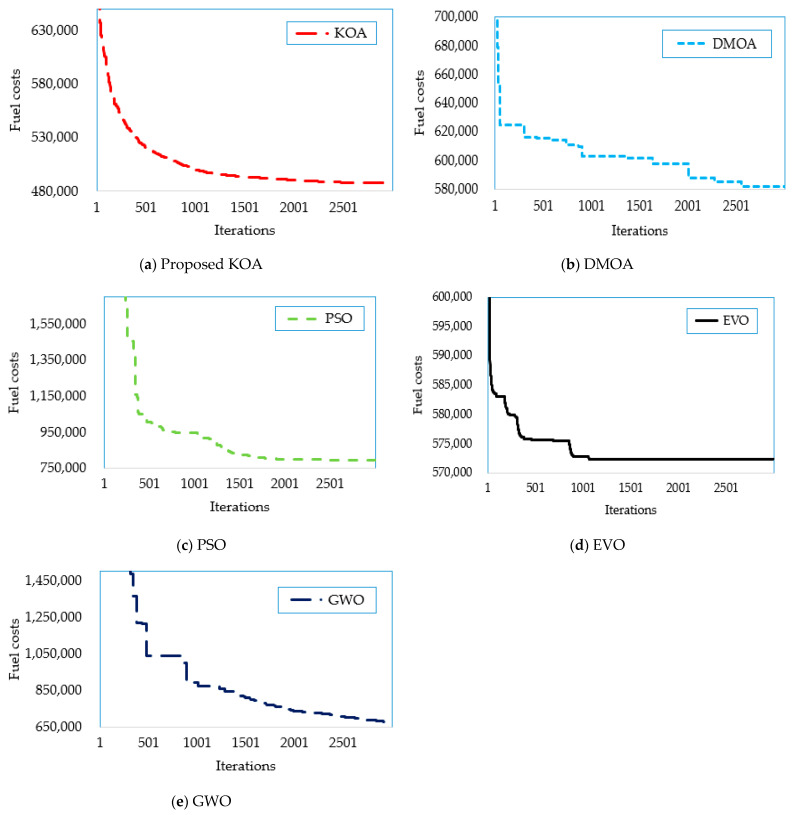
Convergence rates of each individual applied algorithm for the 192-unit CHPUED system.

**Figure 7 biomimetics-08-00608-f007:**
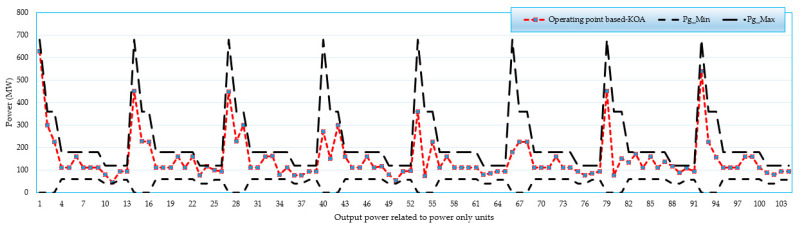
Operating points related to power-only units.

**Figure 8 biomimetics-08-00608-f008:**
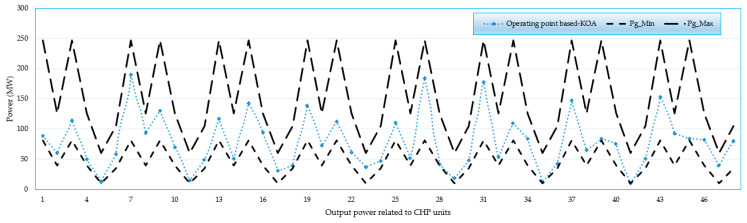
Operating points related to CHP units in terms of their output power.

**Figure 9 biomimetics-08-00608-f009:**
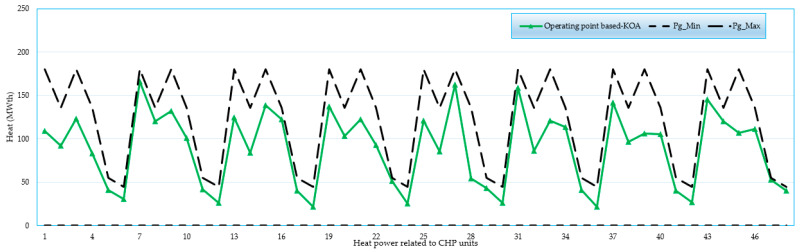
Operating points related to CHP units in terms of their output heat.

**Figure 10 biomimetics-08-00608-f010:**
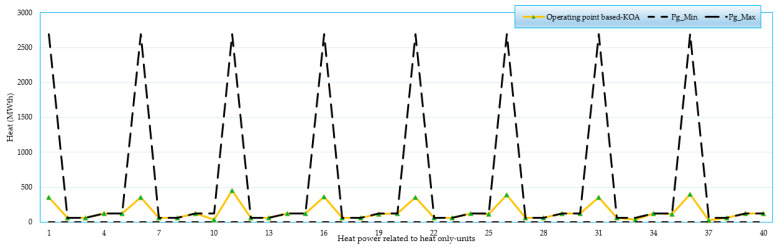
Operating points related to heat-only units.

**Table 1 biomimetics-08-00608-t001:** Test outcomes of all units for the 48-unit test system obtained using the KOA.

Unit	KOA	Unit	KOA	Unit	KOA
Pg1	448.812	Pg22	109.8988	Hg31	40.81127
Pg2	299.5241	Pg23	77.46306	Hg32	26.16548
Pg3	150.1696	Pg24	40.16544	Hg33	111.9561
Pg4	159.738	Pg25	92.93015	Hg34	91.23516
Pg5	109.9199	Pg26	55.37272	Hg35	115.2107
Pg6	159.8738	Pg27	91.53579	Hg36	101.8837
Pg7	110.3709	Pg28	45.4442	Hg37	40.00348
Pg8	159.7404	Pg29	91.34668	Hg38	26.99712
Pg9	109.939	Pg30	53.92644	Hg39	418.7711
Pg10	77.48522	Pg31	11.89511	Hg40	59.99864
Pg11	77.51255	Pg32	48.56369	Hg41	59.99945
Pg12	94.00181	Pg33	93.75923	Hg42	119.9865
Pg13	92.45571	Pg34	58.81029	Hg43	119.9993
Pg14	448.9186	Pg35	99.55169	Hg44	418.9729
Pg15	224.4459	Pg36	71.14285	Hg45	59.99973
Pg16	225.5116	Pg37	10.01567	Hg46	59.99715
Pg17	109.8786	Pg38	50.40061	Hg47	119.9984
Pg18	109.877	Hg27	110.7085	Hg48	119.9969
Pg19	109.9607	Hg28	79.69631	Sum (Pg)	4700
Pg20	159.7711	Hg29	110.5908	Sum (Hg)	2500
Pg21	159.871	Hg30	87.02126	WCTF ($)	116,650.0870

**Table 2 biomimetics-08-00608-t002:** Comparison of the proposed KOA with the reported techniques for 48-unit system.

Optimizer	Min (WCTF ($))	Improving Percentages	Average	Average	Worst	Standard Deviation (Std)
KOA	116,650.0870	-	117,104.5447	117,104.5447	117,915.5359	298.8796
CPSO [19]	120,918.9	3.660%	-	-	-	-
PSO-TVAC [19]	118,962.5	1.982%	-	-	-	-
MRFA [28]	117,336.9	0.589%	-	-	-	-
MVA [28]	117,657.9	0.864%	-	-	-	-
SSA [31]	120,174.1	3.021%	-	-	-	-
MPA[33]	116,860.6	0.180%	-	-	-	-
GSA [29]	119,775.9	2.680%	-	-	-	-
CSA [28]	122,953.5	5.404%	-	-	-	-
MGSO [32]	117,366.09	0.614%	-	-	-	-
DE [28]	120,482.7	3.286%	-	-	-	-
GWO [28]	122,583.3	5.086%	-	-	-	-
CSO and PPS [30]	117,367.09	0.615%	-	-	-	-
JFSO [34]	117,365.09	0.613%	-	-	-	-

**Table 3 biomimetics-08-00608-t003:** Test outcomes of all units for the 96-unit test system obtained using the KOA.

Unit	KOA	Unit	KOA	Unit	KOA	Unit	KOA
Pg1	538.7944	Pg32	110.0436	Pg63	10.02797	Hg70	21.0867
Pg2	299.6272	Pg33	110.3995	Pg64	48.32375	Hg71	109.8662
Pg3	151.2914	Pg34	110.1027	Pg65	82.02374	Hg72	88.07219
Pg4	109.8576	Pg35	159.8276	Pg66	44.18677	Hg73	123.5494
Pg5	109.9343	Pg36	78.60966	Pg67	85.76875	Hg74	99.22525
Pg6	109.8765	Pg37	77.69705	Pg68	73.52597	Hg75	41.11163
Pg7	160.1471	Pg38	57.87178	Pg69	10.29498	Hg76	28.39532
Pg8	110.2941	Pg39	94.20503	Pg70	37.45466	Hg77	408.4901
Pg9	110.342	Pg40	359.9866	Pg71	90.09144	Hg78	59.94804
Pg10	81.05786	Pg41	149.6621	Pg72	55.19146	Hg79	59.99218
Pg11	78.13509	Pg42	149.9737	Pg73	114.497	Hg80	119.9246
Pg12	92.73616	Pg43	110.1571	Pg74	68.07549	Hg81	119.9496
Pg13	92.42638	Pg44	110.1453	Pg75	12.65002	Hg82	410.5949
Pg14	628.4539	Pg45	159.932	Pg76	53.60396	Hg83	59.97848
Pg15	224.6278	Pg46	111.2773	Hg53	124.1457	Hg84	59.97649
Pg16	224.6525	Pg47	160.9166	Hg54	81.36933	Hg85	119.9758
Pg17	109.9961	Pg48	110.255	Hg55	132.9305	Hg86	119.9933
Pg18	159.7635	Pg49	79.55363	Hg56	79.41519	Hg87	405.6305
Pg19	161.739	Pg50	77.98886	Hg57	41.42439	Hg88	58.02127
Pg20	159.8074	Pg51	92.93884	Hg58	31.21828	Hg89	59.99075
Pg21	110.7902	Pg52	92.42203	Hg59	109.8515	Hg90	119.9301
Pg22	110.1921	Pg53	115.5219	Hg60	88.44604	Hg91	119.9778
Pg23	40.25017	Pg54	47.40113	Hg61	131.9541	Hg92	411.7448
Pg24	77.45754	Pg55	131.1481	Hg62	92.84449	Hg93	59.9785
Pg25	93.03079	Pg56	45.14028	Hg63	39.99935	Hg94	59.98367
Pg26	93.02481	Pg57	13.32731	Hg64	26.01246	Hg95	119.9128
Pg27	269.3372	Pg58	59.77202	Hg65	105.3056	Hg96	119.8671
Pg28	224.586	Pg59	90.13865	Hg66	78.57885	Sum (Pg)	9400
Pg29	299.8866	Pg60	55.61345	Hg67	107.3423	Sum (Hg)	5000
Pg30	109.851	Pg61	129.425	Hg68	103.879	WCTF ($)	234,285.3
Pg31	160.1618	Pg62	60.70201	Hg69	40.11558		

**Table 4 biomimetics-08-00608-t004:** Comparison of the proposed KOA with the reported techniques for 96-unit system.

Optimizer	(WCTF ($/h))	Improving Percentages	Average	Worst	Std
KOA	234,285.2584	-	235,683.2917	236,929.2188	761.7006
JFSO [34]	235,277.05	0.423%	236,688.7625	237,940.189	859.1088
HT [34]	235,102.65	0.349%	236,853.3030	239,119.459	1594.7970
HHTJFSO [34]	234,836.04	0.235%	235,646.1289	236,967.064	764.9310
WOA [24]	236,699.15	1.030%	237,431.4678	238,877.049	971.5473
IMPA [33]	235,260.3	0.416%	-	-	-
MPA [33]	236,283.1	0.853%	-	-	-
PSO-TVAC [19]	239,139.5018	2.072%	-	-	-
WVO-PSO [35]	238,005.79	1.588%	-	-	-
WVO [35]	240,861.3210	2.807%	-	-	-
SDO [27]	236,185.18	0.811%	-	-	-

**Table 5 biomimetics-08-00608-t005:** Obtained costs of the KOA, DMOA, EVO, GWO, and PSO for the large 192-unit system.

Algorithm	DMOA	EVO	GWO	PSO	KOA
WCTF ($)	581,798	572,324.8	678,051.9	793,224.8	487,145.2
Improving Percentages	19.43%	17.49%	39.19%	62.83%	-

**Table 6 biomimetics-08-00608-t006:** Setting of parameters for the applied algorithms for the large 192-unit system.

Algorithm	Parameter Settings
KOA	*μ_o_* = 0.1; *γ* = 15; number of solutions = 100; maximum number of iterations = 3000.
DMOA	number of babysitters = 3; number of alpha group = 97; number of scouts = 97; babysitter exchange parameter = 431; alpha female—vocalization = 2; number of solutions = 100; maximum number of iterations = 3000.
EVO	number of solutions = 100; maximum number of iterations = 3000.
GWO	number of solutions = 100; maximum number of iterations = 3000.
PSO	cognitive parameter (*c*1) = 2; social parameter (*c*2) = 2; number of solutions = 100; maximum number of iterations = 3000.

**Table 7 biomimetics-08-00608-t007:** Simulation time of the proposed KOA for all cases.

Case Study	Average Time Per Iteration (Seconds)
48-unit CHPUED system	0.0952
96-unit CHPUED system	0.0981
192-unit CHPUED system	0.1399

**Table 8 biomimetics-08-00608-t008:** Computational complexity of the applied algorithm.

Test Case	Dimension	Number of Solutions	Maximum Number of Iterations	Computational Complexity
48-unit test system	60	100	3000	O (1,800,000)
96-unit test system	120	100	3000	O (3,600,000)
192-unit test system	240	100	3000	O (7,200,000)

## Data Availability

Data are contained within the article.

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
