# Peer review of "Kepler Algorithm for Large-Scale Systems of Economic Dispatch with Heat Optimization"

_biomimetics, 2023, doi:10.3390/biomimetics8080608_

Round 1
Reviewer 1 Report
Comments and Suggestions for Authors
The authors should revised their manuscript based on the comments below:
1) Abstract should show the improvement of the study compared to previous ones by using numerical results in %.
2) Check Typos, such as [1][1], [2][2], ..., in Introduction.
3) Motivation of the study should be stated in the first paragraph in Introduction.
4) The paragraph number in Introduction must be reduced.
5) The literature review is not good, please expand it: More references regarding combine heat and power must be cited. For example,
- conventional and modified Algorithms, such as, Particle swarm optimization versions, Cuckoo search algorithm versions, Bat algorithms, etc., for the conventional problem without renewable energies.
- More constraints and renewable energies to expand the problem: transmisison power grids, solar photovoltaic systems, wind turbines, and so on.
- Newly published algorithms regarding the combined heat and power problem
6) Authors presented "Main points" in Introduction. Please clearly mention Novelty and contribution. Additionally, please explain each point.
7) Please exchange location of Section 2 and Section 3.
8) About Algorithm, authors should explain symbols clearly. Xi, Xs, worst(t), fitk(t); Please clarify one value and one vector for parameters.
9) Please present comaprison and selection techniques for the algorithm.
10) Authors should consider more studies about the solved problem in the paper. Just few algorithms were compared in Table 2, Table 4.
11) Please use a Table to present setting of parameters for algorithms in Figure 5.
12) In Figure 5, why did DMOA and EVO have a line convergence characteristic from the first to the last iteration. It is incredible.
13) Simulation time should be shown for all cases.
14) Author should show shortcomings of the applied algorithm, the study, and suggested work to improve them.
Comments on the Quality of English Language
English should be checked and revised.
Author Response
First of all, the authors would like to thank you and the reviewers for reviewing their paper including exerted efforts and spending time for that issue. The authors have done sincere efforts to revise the paper according to the reviewers' comments. We hope that the attached revised version finds your satisfaction.

Reviewer 2 Report
Comments and Suggestions for Authors
This article reports a noteworthy approach to optimize Unit commitment and economic dispatch model of combined heat and power system. The model proposed in this article aims to reduce the total system dispatching operation cost and promote consumption reduction using the Kepler optimization algorithm (KOA). This manuscript offers a relevant and interesting combination of algorithms (Dwarf Mongoose Optimization Algorithm -DMOA, Energy Valley Optimizer, -EVO, Grey Wolf optimization -GWO, and Particle Swarm Optimization -PSO) to benchmark this optimization.
I have identified a few items to further target the studies’ outcomes and recommendations to translate them into material clarification and potential recommendations for future research.
Line 2. While it is descriptive of the topic, the title doesn’t draw a fair interest to the reader. While it is a novel application of the KOA, the KOA is not “novel” since other works have been published on similar use of the Kepler Algorithm for optimization (see this example from 2015: https://www.sciencedirect.com/science/article/pii/S1319157815000476). While the present manuscript offers a pertinent and new application, it should not say it’s a novel algorithm but rather insists on the differences, validation, and improvements of the Kepler algorithm that were achieved.
Line 27-36. The combined heat and power economic dispatch is a significant issue in power system optimization, and since combined heat and power (CHP) technology often uses waste heat generated in the process of power generation to supply heat to other places, the heat optimization question is especially important here (and mentioned in the title) and should be highlighted in introduction more clearly in the initial context.
Line 88-102. This list looks more like an abstract than an introduction. The conclusions shouldn’t be mentioned, but rather the questions that would be answered by the study behind the article. For example, this sentence reads, “A feasibility study is conducted to demonstrate the superiority and robustness of the proposed KOA.” It should rather be replaced by something that would outline the Feasibility study that is presented in order to “validate accuracy and robustness of the proposed KOA.”
Line 180 and following. Please clarify how the gravitational forces at play in the multi-step process are defined. “Fgi introduces the sun attraction force Xs with any planet Xi,” but Xs or Xi are locations, not gravitational forces. It is unclear in the description but somewhat central in the KOA.
Section 4, Lines 266-308. The results are generally presented clearly, with the exception of Table 2, which could be quite hard to digest for the readers and could be better explained in the text to explain the differences and similarities between the optimizers. The authors should describe (almost) all the data presented in the display items in the test results, which would then better support the main conclusions and make it easier to interpret conclusions later on. Similar comments could be applied to Table 4 (Line 352) and Table 5A and 5B (Line 368-369). The latter (Tables 5A and B are especially difficult to read and should be presented in different subsets or summarized with the detailed table in Appendix). Also, an added “Discussion” section would benefit the reader and better justify the conclusion highlighted in section 5.
Section 5. The conclusion is a good summary of significant concluding remarks consistent with the aims of the study, but the preceding section is not well supportive of the partial conclusions. While data appear relevant, it’s left to the readers to interpret data, and the lack of explanation/discussion does not give the best outlook for the presented findings. It could have been pertinent to add potential alternative explanations from the authors' perspective for some of the results, notably what could motivate future investigation or research.
Comments on the Quality of English LanguageThe language quality is generally clear and sound. Some typos that a quick spell-check could resolve are present.
In the Abstract, Line 8, please avoid repeating the terms “economic dispatch” and a third time “economic” in the same sentence on Line 9. Also, the sentence on Line 11 does feel like a wrong translation and should be rewritten or split into two sentences.
Introduction, Line 33. Please validate which reference, on top of [1], is referred to here (or cut the second [1]). Similar comments for Line 40 and Line 42.
Section 4. Lin 278. Please correct the capital letters usage: “As demonstrated from Ttable 1, tThe best cost value…”
Conclusion, Line 402. Please correct “192 Large test systemS”
Author Response

(The authors gave the same response as above.)

Reviewer 3 Report
Comments and Suggestions for Authors
The paper titled "A novel Kepler algorithm for large-scale systems of economic dispatch with heat optimization" is reviewed and here are the comments that the authors are need to be answered in the revised version.
1. The authors are implemented the ELD and the heat optimization in the combined manner. The authors need to clearly mention the changes that occur in the ELD scheduling while combined with the heat optimization.
2. In the first line of abstract the word "economic dispatch" had written twice....correct it.
3. Literature on ELD is available in bulk quantity, but with combined heat optimization is very limited. This should be highlighted by the authors in the paper.
4. Abstract must be revised and it should highlight the efficacy in implementing the kepler algorithm for combined ELD and heat optimization.
5. Literature must be updated with some recent approaches.
6. The work can be extended by considering the emission dispatch of the thermal units. Authors have to make focus on it.
7. Result analysis is sufficient and moreover the conclusion must address the scope of future expansion of this work.
8. Some language errors are detected in the paper, that needs to be rectified.
Author Response

(The authors gave the same response as above.)

Round 2
Reviewer 1 Report
Comments and Suggestions for Authors
The authors have clarified comments and suggestion well. However, the authors should compare their methods to conventional, modified and improved versions of Cuckoo search, paritcle swarm optimization, Bat algorithms in high impact factors journals like IEEE transaction power system, Neural computing and applications, electrical power and energy systems, electric power system research, Energies, Scientia Iranica, and so on. In addition, they should compare population and iteration, and CPU time. The improvement percentage should be calculated and reported.
For the constraints in transmission networks, if the authors cannot apply them, please explain and put the shortcomings in conclusion and futurwork.
Renewable energies are really essential. Please discuss.
Please clarify the complexity of the applied method.
Discuss more about the challenges and advantages of the study.
Comments on the Quality of English Language
English should be checked and revised.
Author Response
The authors would like to thank you and the reviewers for reviewing their paper including exerted efforts and spending time for that issue. The authors have done sincere efforts to revise the paper according to the reviewers' comments. We hope that the attached revised version finds your satisfaction.

Reviewer 3 Report
Comments and Suggestions for Authors
The paper can be accepted
Author Response
Many thanks prof. for your kind words